# Mercury Sources, Emissions, Distribution and Bioavailability along an Estuarine Gradient under Semiarid Conditions in Northeast Brazil

**DOI:** 10.3390/ijerph192417092

**Published:** 2022-12-19

**Authors:** Victor Lacerda Moura, Luiz Drude de Lacerda

**Affiliations:** Laboratório de Biogeoquímica Costeira, Instituto de Ciências do Mar-LABOMAR, Universidade Federal do Ceará-UFC, Av. da Abolição, 3207, Meireles, Fortaleza CEP 60165-081, CE, Brazil

**Keywords:** mercury, emission factors, speciation, sediments, water, biota, estuary

## Abstract

In the semiarid coast of northeast Brazil, climate change and changes in land use in drainage basins affect river hydrodynamics and hydrochemistry, modifying the estuarine environment and its biogeochemistry and increasing the mobilization of mercury (Hg). This is particularly relevant to the largest semiarid-encroached basin of the region, the Jaguaribe River. Major Hg sources to the Jaguaribe estuary are solid waste disposal, sewage and shrimp farming, the latter emitting effluents directly into the estuary. Total annual emission reaches 300 kg. In that estuary, the distribution of Hg in sediment and suspended particulate matter decreases seaward, whereas dissolved Hg concentrations increase sharply seaward, suggesting higher mobilization at the marine-influenced, mangrove-dominated portion of the estuary, mostly in the dry season. Concentrations of Hg in rooted macrophytes respond to Hg concentrations in sediment, being higher in the fluvial endmember of the estuary, whereas in floating aquatic macrophytes, Hg concentrations followed dissolved Hg concentrations in water and were also higher in the dry season. Animals (fish and crustaceans) also showed higher concentrations and bioaccumulation in the marine-influenced portion of the estuary. The variability of Hg concentrations in plants and sediments agrees with continental sources of Hg. However, Hg fractionation in water and contents in the animals respond to higher Hg availability in the marine-dominated end of the estuary. The results suggest that the impact of anthropogenic sources on Hg bioavailability is modulated by regional and global environmental changes and results from a conjunction of biological, ecological and hydrological characteristics. Finally, increasing aridity due to global warming, observed in northeast Brazil, as well as in other semiarid littorals worldwide, in addition to increased water overuse, augment Hg bioavailability and environmental risk and exposure of the local biota and the tradition of human populations exploiting the estuary’s biological resources.

## 1. Introduction

Heat accumulation in the oceans and sea level rise due to global warming, coupled with decreased river flows by dam construction, accentuated by the reduction in annual rainfall in most semiarid regions, makes marine forcing onto estuarine regions more extreme. Changes in estuarine hydrodynamics result in profound modification of their biogeochemistry, particularly in the mobilization and increase in the bioavailability of pollutants, among these, the highly toxic mercury (Hg). Additionally, accelerated development of economic activities concentrated in the coastal zone is augmenting Hg loads from anthropogenic sources to estuaries. In a scenario where regional and global changes are strengthening, increasing exposure to Hg could be maximized due to significant increases in Hg availability. This higher Hg availability is due to Hg complexation with dissolved organic carbon and an increase in Hg methylation, as suggested by results from other mangrove-dominated estuarine environments [1,2,3], and this also significantly impacts marine biological resources and human exposure risk [4,5,6].

The mobilization and changes in the availability of Hg caused by environmental changes are highly influenced by season, particularly in regions with well-defined rainy and dry periods, such as those under semiarid climates. During the short rainy season, higher river fluxes reduce the residence time of fluvial waters inside the estuary and the export of continental-derived materials, including their pollutant load, to the continental shelf [7]. On the other hand, during the long dry season, small fluvial fluxes and the long residence time of fluvial waters inside the estuary are unable to transport particles from the continental runoff to the sea, accumulating them in the estuary. This is particularly worsened by increasing pollutant emissions from anthropogenic sources [8].

In tropical semiarid regions, such as in northeast Brazil, estuaries are dominated by mangroves that are presently expanding their area following global climate change drivers, mostly sea-level rise and saline intrusion into groundwaters [9,10,11]. These ubiquitous tropical coastal ecosystems promote highly reactive biogeochemical environments that facilitate swift changes in metal speciation and may increase metal bioavailability [1,8,11,12,13]. The influence of mangroves in the Hg biogeodynamics in tropical and subtropical ecosystems has been reported in recent publications, and, irrespective of the geographical location, suggests a similar and consistent pattern. In summary, there is an export of particulate Hg during the short rainy season, whereas in the dry season, particulate Hg accumulates in the estuary and most Hg exported to the sea is soluble reactive Hg [1,6,14,15].

This scenario has been intensified in the past decades due to decreasing annual rainfall, and increasing frequency and length of extended droughts (>3 years long), maximized by river damming in the northeast region of Brazil [16,17,18], and may also be occurring in other semiarid littorals worldwide [14]. This has resulted in the augmenting of Hg concentrations in fish sampled from the marine-influenced portion of estuaries in the semiarid coast of the northeast and southeast Brazil, as well as increasing exposure to Hg for humans inhabiting the lower reaches of these estuaries [19,20,21,22]. For example, Moura and Lacerda [21] reported Hg accumulation rates in shrimps (*Litopennaeus vannamei*) 10-times higher in the lower marine-influenced portion of the Jaguaribe River Estuary compared with individuals sampled in the fluvial end of the estuary, thus suggesting higher bioavailability of Hg in the lower estuary. Costa and Lacerda [19] reported increased exposure risk of the traditional human population in the Jaguaribe estuary due to higher Hg content in local fisheries from the lower Jaguaribe estuary. Azevedo et al. [13] reported increases in total mercury and methylmercury concentrations in fish from the lower Paraíba do Sul River in the semiarid coast of northern Rio de Janeiro in southeastern Brazil due to extended droughts. Other bioindicators, however, need to be studied in conjunction with abiotic compartments and the magnitude of drivers of Hg pollution to further confirm the biogeochemical scenario supposedly created by environmental changes.

Integrative studies on emissions and emission factors of natural and anthropogenic Hg sources, distribution and speciation of Hg along estuarine gradients together with multiple bioindicators are urgently needed. Aquatic macrophytes and animals with different food habits and trophic states are good tools for studying Hg speciation and bioavailability. Different habits (submerged, floating or rooted) in the case of plants and different diets in the case of animals allow the investigation of different uptake pathways of Hg and its bioavailability [21,23,24,25,26,27]. Whereas Hg speciation in water and sediments may also help understand the changes in Hg bioavailability [28]. Taking this into consideration, this work analyzes the Hg concentrations in water, suspended particles and bottom sediments and in aquatic plants of different habits and animals with different ecological characteristics in the dry and rainy periods along the Jaguaribe River Estuary, the largest river entirely enclosed in the semiarid region of northeast Brazil, since the accumulation of Hg in plants and animals is associated with the different life habits that expose them to distinct compartments (water and/or sediments) of the environment and to different Hg fractionation among environmental compartments. Therefore, these organisms are ideal for testing the hydro-biogeochemical scenario proposed in earlier studies resulting from global and regional changes.

## 2. Material and Methods

The Jaguaribe River Estuary, which is fully inserted in a semiarid environment, encompasses three municipalities: Itaiçaba, Aracati and Fortim, totaling nearly 90,000 inhabitants. Major Hg sources to the estuary are the inadequate disposal of urban solid wastes, untreated sewage and diffuse effluents from agriculture and shrimp farming. Formerly published inventories of Hg emissions to the Jaguaribe estuary (Lacerda et al., 2011) are available for the beginning of the 21st century. In this study, they were updated using more recent statistics on emission factors and the dimensions of major sources. These estimates were performed following methodologies previously detailed in Paula Filho et al. [29,30,31].

Sampling campaigns were carried out during the rainy and the dry seasons between 2015 and 2019 in five transversal sections along the Jaguaribe River Estuary, from the fluvial to the marine endmember, where water and sediments were collected in triplicate from at least three points in each section (Figure 1).

Hydrochemical variables were measured in situ in surface waters at about 0.5 m depth, corresponding to about 1/3 of the water column, as follows: dissolved oxygen (YSI 556 probe, YSI Inc., Yellow Springs, USA); water temperature, turbidity and electrical conductivity (Compact-CTD model AST D687; JFE Advantech Co., Ltd., Nishinomiya, Japan); pH (Portable 826 pH-meter; Metrohm AG, Herisau). Water samples were collected manually (gloved), inserting Teflon^®^ 250 mL bottles, counter-current and at 10–20 cm of depth, following clean protocols [35,36]. The bottles were double bagged in plastic bags and kept refrigerated for transport. Upon arrival in the laboratory, the samples were filtered in pre-ashed glass fiber filters (0.7 µ) and acidified with 2 mL concentrated, ultra-pure, Hg-free HNO_3_ to prevent Hg reduction and volatilization and adsorption into surfaces of the storage flasks and maintained in a refrigerator at 4 °C. They were analyzed no longer than 24 h after sampling. The filters used were oven dried at 60 °C for 24 h. This drying procedure results in a negligible loss (<2%) of Hg [37]. The particulate material collected in filters was stored in Petri discs, oven dried at 60 °C for 24 h and stored frozen till analysis. Bottom sediments from the top 0–10 cm layer were collected with a pre-cleaned plastic shovel and double-bagged in plastic bags, and kept under refrigeration until arrival at the laboratory, where they were oven dried (60 °C, till constant weight), powdered, homogenized, preserved in hermetically sealed plastic bottles and stored for subsequent Hg quantification.

The aquatic macrophytes were harvested manually from each station, when available, washed in situ with local river water and then placed in plastic bags and transported in a cold chamber. Plant exemplars were stored in the Prisco Bezerra Herbarium of the Federal University of Ceará, Fortaleza, Brazil. All plant samples from each point were composed (*n* = 5) for better representativeness of the environment and were identified to the lowest possible taxonomic level. The following species were sampled, floating plant species: the water hyacinth *Eichhornia crassipes* (Potenderiaceae), rooted submerged species: the Brazilian waterweed *Egeria* sp. (Hyrocharitaceae) and the coontail *Ceratophyllum* sp. (Ceratophyllaceae), rooted emergent: the grasses *Paspalidium paludivagum*, *Blutaparon portulacoides*, *Paspalum* sp. and *Panicum emergens*. Samples were oven dried (60 °C, till constant weight), ground with a porcelain pestle and mortar to avoid contamination and stored in pre-cleaned plastic containers until analysis. Selected animals occurring along the entire estuary, except sessile or of small mobility, were also collected from each station when available. Animal species comprised the white shrimp *Littopenaeus vannamei* and fishes *Eugerres brasilianus* and *Cathorops spixii*. Animals were frozen upon sampling, and subsamples of muscle tissues were lyophilized and stored in glass vials until analysis.

Total dissolved Hg (Diss-Hg) was quantified in a Merlin PSA cold vapor atomic fluorescence spectrometer (CV-AFS) after oxidation with 0.2 mL of a bromine monochloride solution (0.1 mL KBrO_3_ 1% *m/v* + 0.1 mL HCl 20% *v/v*) at room temperature. After oxidation, 0.1 mL of 1% *m/v* ascorbic acid solution was used to reduce the excess BrCl [36]. This was followed by a reduction with 10% SnCl_2_ solution used for reducing the reactive Hg fraction [36]. The detection limit, estimated as three times the standard deviation of reagent blanks, was 0.18 ng·L^−1^ in water. In all cases, the blank signals were lower than 4% of the respective sample’s Hg concentration.

Digestion of dried plants, of lyophilized animal tissues, suspended material and bottom sediment samples was performed similarly, in duplicate. Approximately 0.5 g were weighed in Teflon^®^ tubes and pre-digested for 1 h, at room temperature, with 10 mL of concentrated HNO_3_. The digestion was performed in a microwave oven digester (MARS XPRESS, CEM Corporation) at 200 °C for 30 min. Then, 1 mL of H_2_O_2_ was added, and the final extract was transferred quantitatively in a volumetric flask and brought to a final volume of 100 mL with MilliQ water. All glassware and materials used were previously washed in a neutral detergent bath followed by immersion for 24 h in a Hg-free HCl 10% solution. Quantification was performed by cold vapor atomic absorption spectrophotometer (CV-AAS, NIPON^®^ NIC RA-3) after Hg reduction with NaBH_4_. Certified reference standards: dried leaves of aquatic plant *Lagarosiphon major* (BCR 060 with 340 ± 40 ng g^−1^ of Hg), animal (ERM BB422 with 601 ng g^−1^ of Hg) and estuarine sediment (NIST 1646A with 40 ng g^−1^ of Hg) were simultaneously analyzed to evaluate the accuracy and recovery of the method. The recovery for reference materials was averaged, resulting in 88.4 ± 6.6% for aquatic plants, 88.4 ± 6.6% for sediments and 101.7 ± 6.3% for animals; the detection limit of the procedure was 0.02 ng g^−1^, and the quantification limit was 0.06 ng g^−1^. The concentration values were not corrected for the relative recoveries obtained for the certified materials.

The normality and the variance homogeneity of the data were tested using the Shapiro–Wilk test. Variance Analysis (ANOVA) and Kruskall–Wallis test was used to compare the concentrations of Hg in plant parts and the Hg concentrations in water, sediment and animals along the estuary. Bioaccumulation curves were designed, and the Pearson correlation coefficients were applied to observe the relationship between Hg content in animals with their size. These bioaccumulation curves and the *Mann–Whitney* test were also used to recognize contrasts between Hg concentrations in animals collected in the marine and fluvial end members. Comparisons between seasons were made through *t* and Mann–Whitney tests. The significance value used for the tests was 95% (*p* < 0.05). Statistical tests and the preparation of graphs were performed with Past 3.05 (Copyright Hammer 1999–2015) and Microsoft^®^ Office 2016 (Microsoft Corporation 2016).

## 3. Results and Discussion

### 3.1. Sources and Emissions of Hg

Major sources of Hg to the Jaguaribe River Estuary were first evaluated in 2005 [38] using estimated emission factors from major sources and their magnitude. In this study, emissions from major sources were updated using the more recent available statistics on urbanization, industries and agriculture occurring in the estuarine basin, also using more recent statistics on emission factors and the dimensions of major sources. These estimates were performed following methodologies previously detailed in Paula Filho et al. [29,30] and Oliveira et al. [3] and were much more precise than those used in 2005. In summary, industrial sources are still inexistent in the estuarine and lower Jaguaribe river basin, as was in 2005, and are not considered here. The human population of the three main municipalities increased by 8% relative to 2005, and this increase was considered when calculating emissions by urban sources, which include solid waste disposal and wastewaters. No significant changes in disposal or treatment procedures have occurred in the past 15 years, therefore, emission factors are likely the same as in 2005, and only the correction to the new population data was used to estimate loads from untreated wastewater and solid urban wastes. There was, however, a huge expansion of shrimp aquaculture, a source presenting very high emission factors for Hg [6,39]. Updated emission factors of Hg from these three major Hg sources to the Jaguaribe estuary were estimated as 200 mgHg ha^−1^ year^−1^ for urban wastewater, 400 mgHg ha^−1^ year^−1^ for solid waste disposal and 374 mgHg ha^−1^ year^−1^ for shrimp aquaculture. Considering the present-day magnitude of major sources and their relative emission factors, the estimated annual Hg emissions to the estuarine basin reach 81 kgHg year^−1^ for wastewater, 162 kgHg year^−1^ for solid wastes disposal and 0.35 kgHg year^−1^ from intensive shrimp aquaculture, summing up to a total annual Hg load to the basin of approximately 243 kg.

### 3.2. Hydrochemistry

The hydrology and hydrochemistry of the Jaguaribe River Estuary have been monitored by numerous studies since the initial works by Marins et al. [40,41]. Salinity, temperature and dissolved oxygen observed in the sampling campaigns fall within the reported range of values for this estuarine environment. In summary, the water temperature is high and has been quite constant throughout the year and along the past two decades of monitoring. Temperature diuturnal variation is around 1.5 °C between neap (30.5 °C) and spring (29.0 °C) tides. Year-round temperature variation is slightly larger, from 28 to 31 °C. A larger difference in other state parameters, particularly salinity, is observed along the estuary during extreme drought and flood events, when salinity may reach 29 in the fluvial endmember at Itaiçaba in the dry season, while in extremely rainy years, this station registers very low salinity (<0.1). Along most of the estuary, however, salinity typically varies tidally between 5 and 35. The estuary’s main channel is fairly oxic with dissolved oxygen generally >5 mg L^−1^. Exceptions are tidal creeks dominated by mangroves and those receiving shrimp farm effluents, where oxygen levels can drop to values <0.5 mg L^−1^ [7,8,33,34,40,41,42,43,44,45,46].

### 3.3. Mercury in Sediments and Water

Average Hg concentrations in bottom sediments were significantly higher (9.94 ng g^−1^) in the dry period compared to the rainy season (6.14 ng g^−1^). On the other hand, average particulate Hg (Part-Hg) concentrations were higher in the rainy season (7.46 ng L^−1^) than in the dry season (1.10 ng L^−1^), while average dissolved Hg (Diss-Hg) concentrations were significantly lower in the rainy season (1.12 ng L^−1^) than in the dry season (12.32 n L^−1^) (Table 1). This seasonal distribution pattern of Hg concentrations agrees with previously estimated balances and modeling of Hg in the Jaguaribe River Estuary, suggesting accumulation in sediments within the estuary and the release of dissolved Hg during the dry season and exportation to the sea, whereas the export of particulate Hg occurs only in the rainy season [6,8]. The relatively lower Hg concentrations in sediments may result from erosion of the surface Hg-richer sediments during the rainy season, this also results in higher Hg concentrations in the particulate form during the rainy season. However, the origin of this Hg may also include new seasonal emissions from stronger surface runoff from the upstream basin.

Figure 2 shows the spatial distribution of Hg in abiotic compartments along the estuarine gradient. Sediment Hg concentrations were higher in all stations in the dry season but varied little. Relatively higher concentrations are observed in the middle estuary (Cabreiro and Vila Volta stations, Figure 1).

Part-Hg in the rainy season decreases from the river endmember to the middle estuary, suggesting erosion and transport of particles from the upstream basin, and increases thereafter toward the sea, suggesting the remobilization of surface bottom sediments from the estuary. In the dry season, Part-Hg is very low and varies little throughout the estuary, suggesting more efficient particle settling facilitated by the higher residence time of the water mass [6,8,40]. Diss-Hg in the rainy season is low throughout the estuary but increases slightly from the fluvial endmember to the middle estuary and then decreases toward the sea. In the dry season, however, Diss-Hg increases continuously from the fluvial endmember to the sea, with the highest concentrations in the marine endmember of the estuary (Fortim station, Figure 1).

Previous studies in the Jaguaribe River obtained similar results to those found in this study regarding the concentration of Hg in sediments and waters [8,19,20,48]. Marins et al. [41] also observed that, during the rainy season, there is a small variation of Hg contents in bottom sediments throughout the estuary. Lacerda et al. [8] associated higher Diss-Hg concentrations and fluxes in the dry season with extremely low rainfall and the operation of dams, reducing sediment transport and augmenting the residence time of the fluvial water mass in the estuary, increasing the reactivity of Hg, which includes the most bioavailable organic Hg forms, favoring the incorporation by the biota and biomagnification in the food chain. Extensive and expanding mangroves favor this by producing large amounts of dissolved organic matter [49] with a large Hg complexing capacity, as observed previously in the Jaguaribe estuary and in other mangrove-dominated semiarid estuaries [6,40,50].

Higher Diss-Hg and lower Part-Hg concentrations in dry periods relative to rainy periods have been reported in many estuaries, mostly in regions with sharp differences in rainfall and, therefore, in river flow, between seasons, e.g., Coulibaly et al. [51] in Biétri Bay, Ivory Coast, higher mercury concentrations were recorded during the dry season, Saniewska et al. [12] in the Arctic coast, Cardoso et al. [52] in Ria do Aveiro, southern Portugal, Azevedo et al. [13] in the Rio Paraíba do Sul estuary on the dry coast of northern Rio de Janeiro. In China, the overuse of water resources in the middle and higher reaches of several rivers results in less and less water flowing into estuaries. Consequently, some recent studies have also observed increasing seasonal differences in Hg partitioning along Chinese estuaries due to excess upstream water withdraw (Tong et al. [26], Jiang et al. [53]). Therefore, increasing mobilization and bioavailability of Hg in estuaries has been highlighted as a ubiquitous, environmentally significant response to increased drought because of water resource overuse, reduced annual rainfall and increased frequency and duration of extended droughts caused by global climate change [6].

### 3.4. Mercury in Aquatic Macrophytes

Aquatic species of plants were not evenly distributed along the estuary, except for the emergent ones, which occurred at all five stations. The floating *E. crassipes* and the submerged *Ceratophyllum* sp. occurred only in the fluvial endmember (upstream and downstream Itaiçaba, Figure 1). Therefore, these species were only used to distinguish between seasonal Hg concentrations in water and sediments rather than Hg concentrations changes along the estuarine gradient.

Concentrations of Hg in the floating *E. crassipes* and the submerged *Ceratophylum* sp. in the fluvial stations and in the dry and rainy seasons are shown in Figure 3. As for seasonal variability of the Hg content, the results are similar for the two species, with dry season Hg conservations higher than in the rainy season. Both fluvial stations presented similar concentrations in all plant tissues. The highest concentrations were observed in roots, followed by leaves and stems. Higher Hg concentrations occurred in the dry season in the three tissues of *E. crassipes* and *Ceratophyllum* sp. compared to the rainy season.

Figure 4 shows the ratio between Hg concentrations measured in plant tissues from all sampled species in the dry and wet seasons. The seasonal enrichment of Hg that occurred in all tissues and plant habitats was particularly high (>3.0) in the stems and leaves of emergent macrophytes. The seasonal variation in Hg concentrations in plants agrees with the variability of dissolved Hg. In general, floating and submerged macrophytes efficiently uptake Hg through rhizofiltration and further translocate and accumulate it in the other plant tissues. However, translocation rates from roots to other leaves are, in general, smaller than 1.0 [23]. Root/leaf ratios were similar between dry (0.63) and rainy (0.53) seasons, strongly suggesting that seasonal changes in Hg contents are due to changes in environmental levels rather than different seasonal physiological metabolism of the plant. The strong relationship between Hg concentrations in floating plants and water has been the basis for using this organism to clean up contaminated aquatic ecosystems, as reported in other studies [23,54,55].

Rooted, emergent macrophyte *Paspalum* spp. showed the highest concentration of Hg in roots and, contrary to floating and submerged plants, presented the highest Hg concentrations during the rainy season, not responding to any increase in sediment concentrations or the availability of Hg in water (Figure 5); although the response of rooted aquatic macrophytes to Hg contents in sediments has been observed in many estuaries, and highest concentrations have also been observed in roots [56,57]. More detailed studies are needed to better understand the variability of Hg in the roots of this species. Notwithstanding, there is, in general, an increase in Hg content in all plant tissues from the fluvial to the marine endmember of the estuary, suggesting an increase in Hg availability to plant uptake.

In summary, this pattern of Hg distribution between organs and between plants of different habitats is in accordance with previous observations on Hg distribution in aquatic macrophytes; these plants have different capacities for the removal of trace metals in different plant organs [54]. In most cases, metals are more concentrated in plant roots, as they are considered more important for the absorption of Hg than leaves and stems [24,57], and this is particularly true in rooted emergent macrophytes. Aquatic macrophyte species that accumulate a higher concentration of metals in the root than leaves are considered more tolerant. This means that they cannot avoid the absorption of these elements but limit their translocation [57,58]. For this reason, these plants are considered potentially phytostabilizing, retaining Hg in the soil and avoiding its expansion to adjacent areas, since about 80% of the Hg burden in higher plants remains in root walls forming insoluble precipitates with anionic compounds [58]. Some species, particularly floating and submerged species, can absorb metals by roots and rhizomes, as well as by leaves [57]. Lafabrie et al. [23] found the capture and accumulation of Hg by some plants through water, highlighting that the capture of Hg can occur from the water column, via leaves, stems and sediments, by the roots [23].

Our results, so far, demonstrate that floating and submerged species respond to the availability of Hg in the water column, as suggested by the higher Hg content in the dry season, whereas rooted emergent species (*Paspalum* spp.) respond to Hg in sediments, and thus show highest Hg concentrations in the rainy season.

### 3.5. Mercury in Aquatic Fauna

The distribution of Hg in animals, due to logistical problems, did not cover the two seasonal periods; thus our discussion will be based on the spatial distribution only. The distribution of Hg in water and sediments along the estuarine gradient suggests higher Hg concentrations in animals found in the lower and middle estuary compared to the upper estuary. Figure 6 displays differences in Hg concentrations in fish *Eugerres brasilianus* and *Cathorops spixii* and shrimps *Litopannaeu vannamei* sampled from the two portions of the estuary, areas dominated by fluvial (FDA) and marine (MDA) conditions.

Significant differences were observed between the areas, with fish and shrimps presenting higher Hg concentrations in the marine-influenced portion of the estuary. Average Hg concentrations in the fish *C. spixii* varied from 49.2 ± 3.2 ng·g^−1^ in the fluvial endmember to 101.3 ± 10.7 ng·g^−1^ in the marine endmember, whereas in the fish *E. brasilianus* varied they from 30.8 ± 1.2 ng·g^−1^ in the fluvial endmember to 59.4 ± 7.2 ng·g^−1^ in the marine endmember. Average Hg concentrations in the shrimp *L. vannamei* varied from 10.7 ± 0.5 ng·g^−1^ in the fluvial endmember to 13.3 ± 1.3 ng·g^−1^ in the marine endmember. Fish Hg concentrations are over 10-times higher than in shrimps. These differences are associated with diet, mostly carnivorous fish and detritivores shrimps, and the lifes-pan of months to a few years of fish to a few weeks of shrimps.

Costa and Lacerda [20] also observed higher Hg concentrations in different fish species sampled in the marine portion of the Jaguaribe estuary compared to those sampled in the fluvial-dominated portion and these higher concentrations resulted in a three-times increase in exposure to Hg of the local traditional fishermen populations inhabiting the river margins. Moura and Lacerda [21], comparing the same portions of the Jaguaribe estuary, observed significantly higher Hg concentrations in fishes and invertebrates sampled in the marine-influenced portion. In addition, the authors noted an increase in the Hg bioaccumulation rate in the marine endmember species, mainly in the white shrimp *L. vannamei*. These results suggest a relationship with major environment conditions that differ in the two estuarine portions derived from the longer residence time of the water mass, which promotes longer periods of chemical interactions among substances present in the water and results in the formation of organo-metal complexes with higher reactivity of metals [1,8,50]. This difference is strengthened by the scarce rainfall and the presence of dams in river basins of semiarid regions reduce water discharge to the coast and, in conjunction with tidal forcing, increases the water residence time in the estuary from about 0.2 to up to 13 days, triggering accelerated sedimentation of suspended particles, and colonization by mangroves, increasing the availability of dissolved organic compounds that strongly bind Hg, increasing its bioavailability. This chain reaction is stronger during extended dry periods. In addition, global climate change is strengthening these reactions [6,8,14,33,44]. Unfortunately, in the present work, we could not study the eventual seasonal impact on Hg concentrations in animals, such as it has been evidenced for plants. However, this relationship between dry periods and Hg bioavailability has already been described by Barletta et al. [59], reporting total Hg content in *Trichiurus lepturus* decreasing with increasing rainfall when most Hg entering the systems were low-bioavailable particulate Hg. Further, Azevedo et al. [13] observed increases in Hg concentrations in three fish species in Paraíba do Sul River in southeastern Brazil during a long-term drought period.

The large amount of dissolved organic carbon (DOC) from mangrove sulfate reduction in marine-dominated estuarine areas also contributes to methylmercury (MeHg) production [1,15,60,61], a Hg species with high bioavailability and toxicity, in addition, Mounier et al. [50] demonstrated the young and labile nature of this DOPC formed in the mangrove-dominated Jaguaribe estuary, and this can explain the elevated Hg concentrations in species from marine-dominated areas. The spatial distribution of Hg concentrations in *E. brasilianus*, a low mobility species but the only species that we could sample along the entire Jaguaribe River Estuary (Figure 7), strongly suggest this, with the highest values observed in Fortim City (97 ± 27 ng·g^−1^), followed by Vila Volta (57 ± 6 ng·g^−1^), Jardim (51 ± 13 ng·g^−1^), Aracati (33 ± 2 ng·g^−1^), Cabreiro (30 ± 2 ng·g^−1^) and Itaiçaba (30 ± 3 ng·g^−1^).

Biological factors such as age, length and weight strongly contribute to increases in Hg concentrations in aquatic organisms [62,63,64,65]. Therefore, merely comparing Hg concentrations from different sites in a gradient may not be significant if these variables are not uniform throughout. Rather, bioaccumulation curves relating to one of those variables with Hg concentrations are compared for a given species at different sites of a gradient could provide a better understanding of the impact environmental variables play on Hg contents. In Figure 8, we can observe that all three species presented higher Hg bioaccumulation in marine-dominated areas than in fluvial. These results show different bioaccumulation rates between the sampled areas, presumably the influence of the Hg available due to semiarid conditions, as we presented earlier, cause this difference. The detritivore species *L. vannamei* showed the highest distinction in bioaccumulation rate, probably a result of its shorter lifespan relative to the fish species. Besides the elevated size of fluvial organisms, the highest concentrations were found in marine individuals and presented an accumulation rate over one order of magnitude higher than organisms collected in fluvial areas.

## 4. Conclusions

The monitoring of hydrochemistry and Hg content in different compartments of the Jaguaribe estuary showed a distinct Hg dynamic between the two major portions of the estuary due to typical environmental settings of the semiarid region that mainly affects the biota of marine-influenced areas of the estuary, where the Hg deposition and reactivity with dissolved organic matter results in relatively higher concentrations. The initial hydrochemical studies characterize significant differences between drought and rainy periods, contributing to changes in Hg concentrations in water and sediments. The longer retention of water mass in the estuary during drought periods affects the sediment transport to the continental shelf, unleashing a high Hg concentration from the estuary sediment, as suggested by the higher Hg concentrations in sediments in the dry period relative to the rainy season. In water, the Hg dissolved fraction, constituted by high bioavailable species, exhibited high content during the dry season and floating and submerged plants responded with higher Hg concentrations.

Besides the scarcity of data to compare fishes and shrimp species between seasons, all three species showed a similar pattern of bioaccumulation seen in floating and submerged plants along the estuarine gradient. Species showed an increase in Hg concentrations downstream, with higher contents found in the marine-influenced portion of the estuary. In addition, bioaccumulation rates were also higher in individuals captured in the marine-influenced portion, confirming higher Hg bioavailability there.

Finally, the major drivers responsible for the increase in Hg mobility and bioavailability in estuaries of semiarid regions are strongly affected by global climate change. Decreasing rainfall and increasing ocean forcing will contribute to accelerating Hg cycling, which, consequently, will increase the contamination of the biota and human exposure to this highly toxic pollutant.

## Figures and Tables

**Figure 1 ijerph-19-17092-f001:**
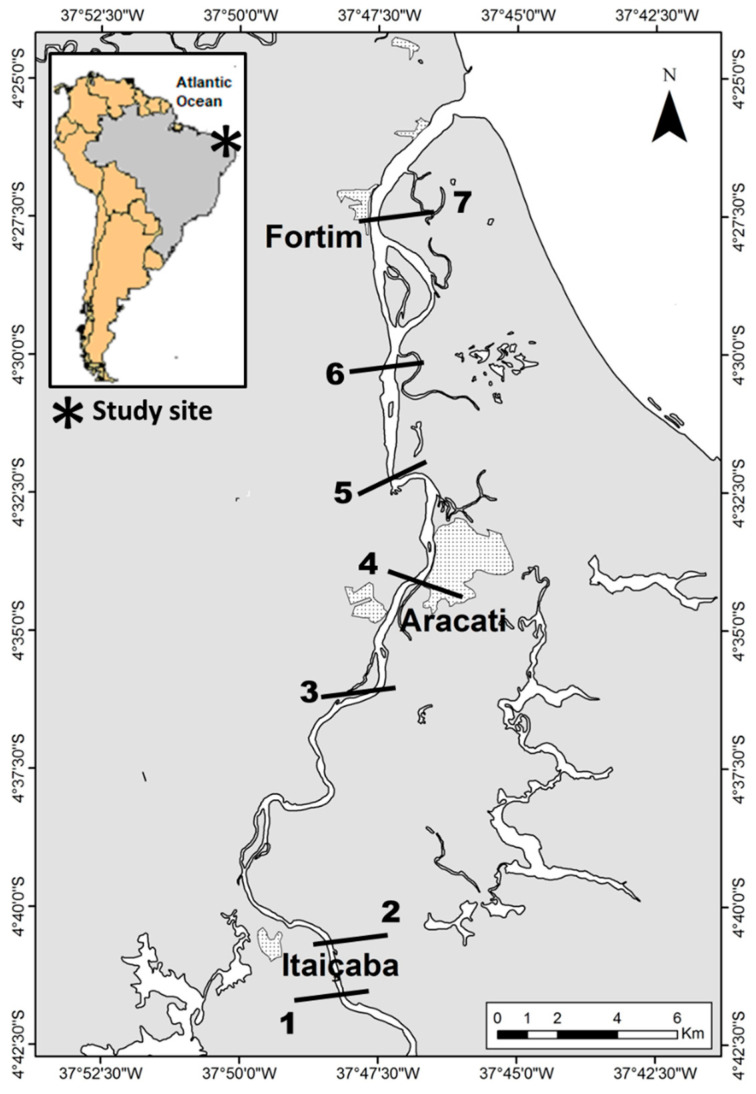
Map showing municipalities and the location of sampling stations along the Jaguaribe River Estuary in northeast Brazil. 1. Upstream Itaiçaba. 2. Downstream Itaiçaba. 3. Cabreiro. 4. Aracati City. 5. Vila Volta. 6. Jardim. 7. Fortim City. Sampling stations were chosen according to river sections defined based on major physical–chemical variables by Dias et al. [7,32,33,34].

**Figure 2 ijerph-19-17092-f002:**
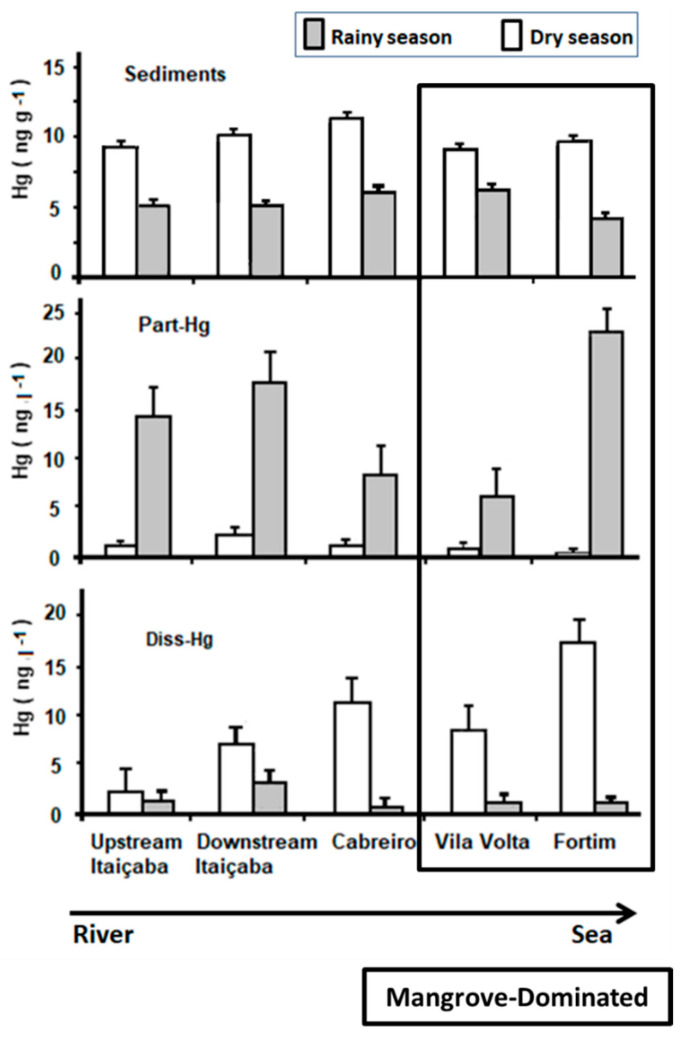
Mercury distribution in water, suspended particles and bottom sediments along the estuarine gradient of the Jaguaribe river in northeast Brazil. Station locations are shown in Figure 1. Vila volta and Fortim stations have been characterized as the middle estuary according to state variables measured by previous long-term monitoring [7,33,34,43,44,47].

**Figure 3 ijerph-19-17092-f003:**
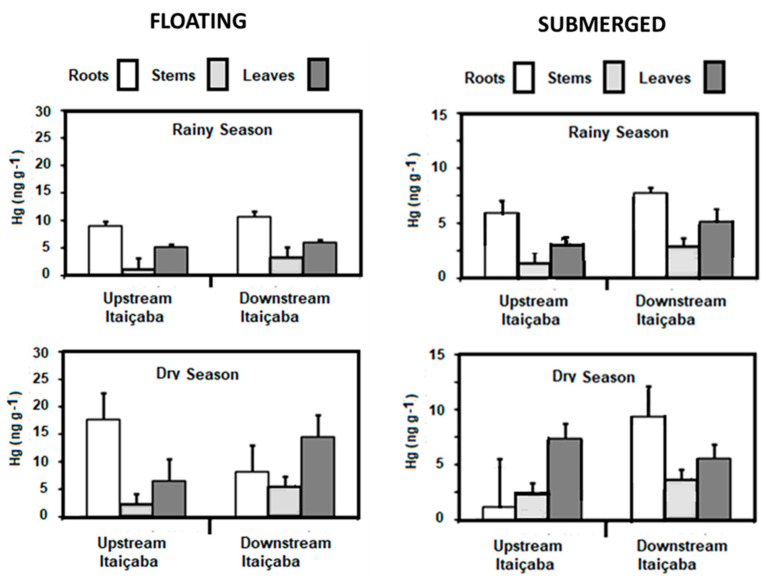
Seasonal distribution of Hg concentrations in floating (*E. crassipes*) and submerged (*Ceratophyllum* sp.) aquatic macrophytes in the fluvial stations of the Jaguaribe River Estuary.

**Figure 4 ijerph-19-17092-f004:**
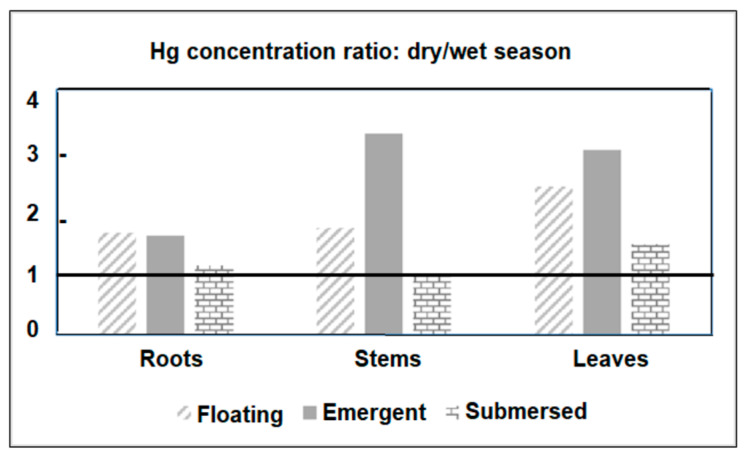
Ratios between Hg concentrations observed in different organs of plants from different habits in the Jaguaribe River Estuary in northeast Brazil.

**Figure 5 ijerph-19-17092-f005:**
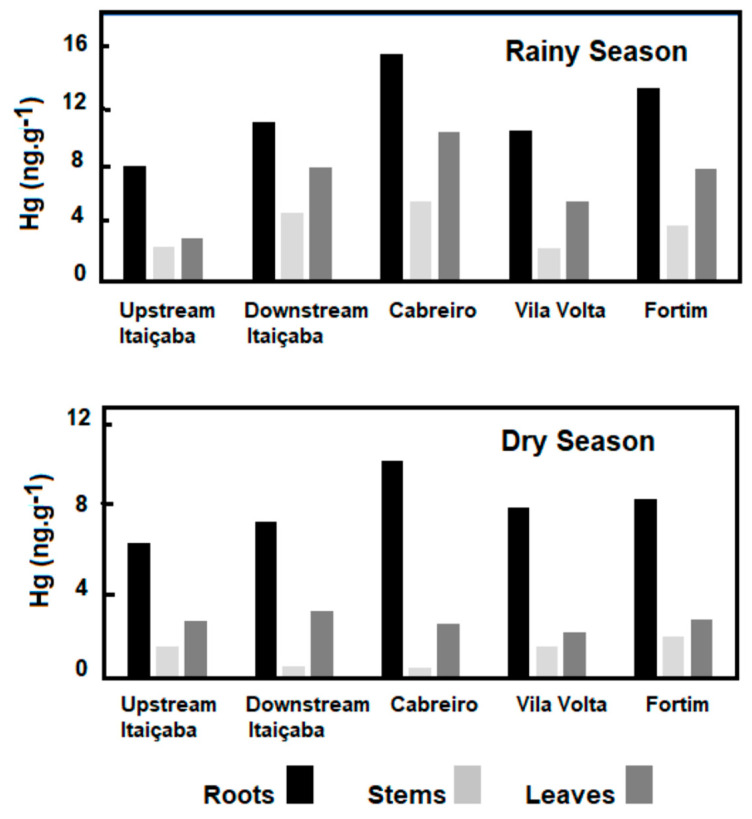
Spatial distribution of Hg concentrations along the estuary gradient of the Jaguaribe River Estuary in the dry and rainy seasons in the rooted emergent macrophyte *Paspalum* spp.

**Figure 6 ijerph-19-17092-f006:**
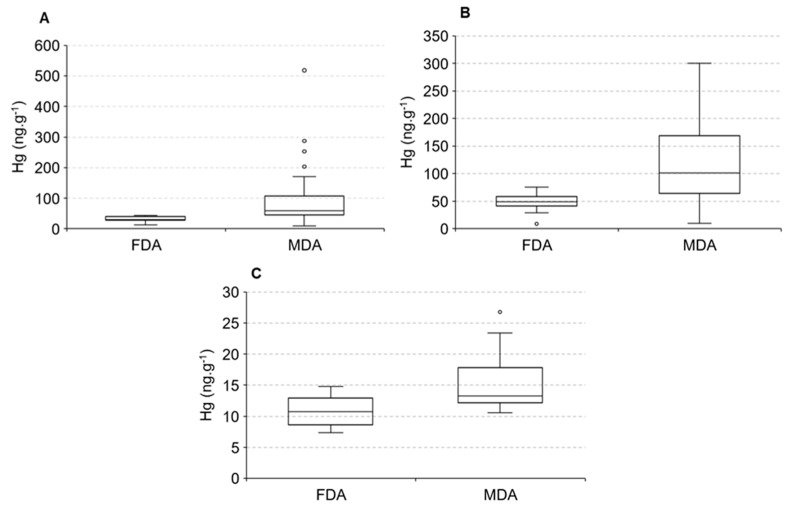
Mercury concentrations in fluvial-dominated areas (FDA) and marine-dominated areas (MDA) in *Eugerres brasilianus* (**A**), *Cathorops spixii* (**B**) and *Litopannaeu vannamei* (**C**).

**Figure 7 ijerph-19-17092-f007:**
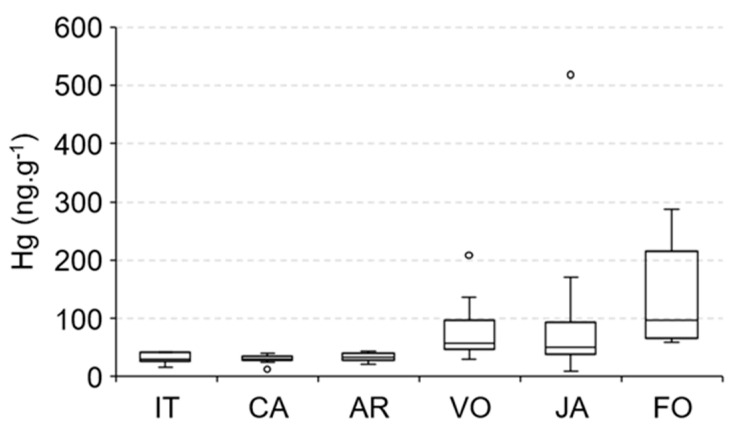
Spatial distribution of Hg concentrations in the fish *E. brasilianus* along the Jaguaribe River Estuary: IT—Upstream and downstream Itaiçaba, CA—Cabreiro, AR—Aracati City, (fluvial influenced) VO—Vila Volta, JA—Jardim, FO—Fortim City (marine influenced).

**Figure 8 ijerph-19-17092-f008:**
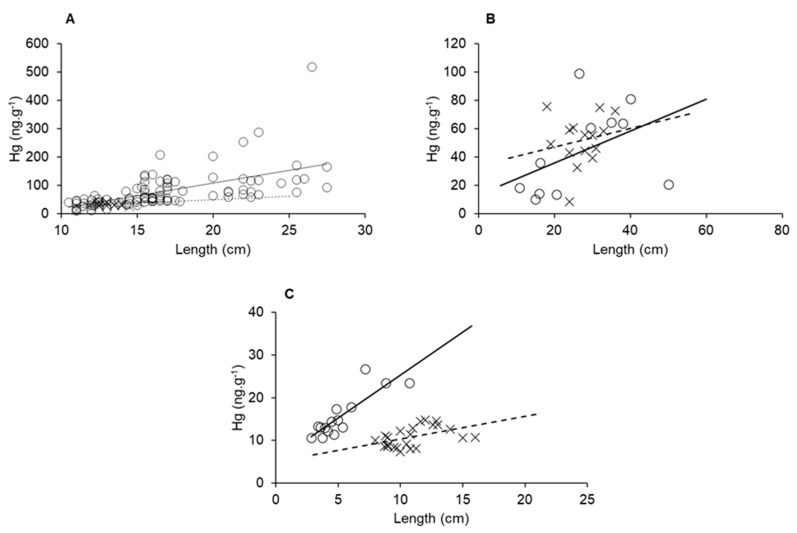
Size and Hg bioaccumulation in *E. brasilianus* (**A**), *C. spixii* (**B**) and *L. vannamei* (**C**) sampled in a fluvial-dominated area (x point, dashed line) and a marine-dominated area (circle point, continuous line).

**Table 1 ijerph-19-17092-t001:** Seasonal distribution of Hg concentrations (ng g^−1^ for sediments and ng L^−1^ for dissolved and particulate Hg) in the dry and rainy seasons in the Jaguaribe River Estuary, considering all sampling stations (*n* = 15). * Significantly different at *p* < 0.05.

	Sediments	Diss-Hg	Part-Hg
**Rainy season**	6.14 ± 0.41	1.12 ± 0.28	7.46 ± 2.07
**Dry season**	9.94 ± 0.40	12.32 ± 2.63	1.10 ± 0.44
**Student’s *t p*-Value**	0.001 *	0.003 *	0.017 *

## Data Availability

Original datasets are available upon request to the corresponding author (ldrude1956@gmail.com).

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
