# Peer review of "Mercury Sources, Emissions, Distribution and Bioavailability along an Estuarine Gradient under Semiarid Conditions in Northeast Brazil"

_ijerph, 2022, doi:10.3390/ijerph192417092_

Round 1

Reviewer 1 Report

Please find my comments in attach

Author Response

Please find attached the response to reviewer 1

Reviewer 2 Report

The topic is very interesting and close to public focus. The manuscript is showing the mercury sources, emissions, distribution and bioavailability along an estuarine gradient under semiarid conditions in NE Brazil, which is a case study for public health and relevant to global environment.

Several questions:

1.     Would you please include the rule for locations of five transversal sections along the Jaguaribe River estuary?

2.     NE/northeastern and so on, please be consistent.

3.     L130, L134, L146, and L222, L227, L228 check and improve.

4.     Would you please do a comparison of major source evaluation between the previous evaluation (Lac- 193 erda and Sena, 2005) and new employed in this study? And let readers know why you employ the updated one/New

5.     L193-215, do we have any available figure to show clearly your results instead of only words? Which would help readers know your evaluation better?

6.     L217-232, also should have figure or diagram for your wordings?

7.     Figure quality should be carefully improved.

8.     “impact of anthropogenic sources on Hg bioavailability is modulated by regional and global environmental changes and results form a conjunction of biological, ecological and hydrological characteristics” I would like to see this more reliable or direct from your new evaluation data.

9.     The reference should not be included in the conclusion part? Where you should include your major conclusions through your study? Please reorganize your conclusion

Author Response

Please find attached the response to reviewer 2

Reviewer 3 Report

This study investigated the source, emissions, distribution of Hg along semiarid coast of northeastern Brazil. This work is well organized and easy to understand. Only a few questions need to be solved before getting published.

1. The novelty of this work should be clearly restated.

2. The language should be checked and polished by native speakers.

3 The abstract and conclusion are quite long, please refine them.

Author Response

Please find attached the response to reviewer 3

Round 2

Reviewer 1 Report

The manuscript has been improved from the previous version and in my opinion can be considered for publication in IJERPH after spell check (for example, in line 339 of revised version: remove the "and" before (Ceratophyllum sp.)).

Moreover, the discussion about Hg concentrations in Paspalum spp. in lines 359-362 in my opinion is still a bit confusing: it is reported that Hg contents in these plants are higher in rainy then dry season "responding to the increased sediment concentrations rather than the availability of Hg in water", but in lines 254-255 it is reported that "Average Hg concentrations in bottom sediments were significantly higher (9.94 ng.g−1) in the dry period compared to the rainy season (6.14 ng.g−1)." Is the word "increased" in line 361 referred to the natural background Hg concentrations in sediments rather than to the seasonal trend? If so it should be better specified

Author Response

- ... in line 339 of revised version: remove the "and" before (Ceratophyllum sp.)): Removed

- ... the reviewer is right, there is no relationship betwween root Hg and sediemnt or water Hg contents. We have rewritten the paragraph entirely, since we can not explain based on our resuts only the higher Hg in roots during the reainy season.

"Rooted, emergent macrophytes Paspalum spp. showed the highest concentrations of Hg in roots and contrary to floating and submersed plants, presented the highest Hg concentrations during the rainy season, not responding to any increase in sediment concentrations or the availability of Hg in water (Figure 5); although the response of rooted aquatic macrophytes to Hg contents in sediments has been observed in many estuaries, and highest concentrations have also been observed in roots (Marins et al., 1997, Chen and Yang, 2012). More detailed studies are needed to better understand the variability of Hg in roots of this species. "

Hope we have reached the necessary understanding relating these questions.

Sincerely yours,

Luiz Ddrude de Lacerda
